# Inhibition of Viral Membrane Fusion by Peptides and Approaches to Peptide Design

**DOI:** 10.3390/pathogens10121599

**Published:** 2021-12-09

**Authors:** Nejat Düzgüneş, Narcis Fernandez-Fuentes, Krystyna Konopka

**Affiliations:** 1Department of Biomedical Sciences, Arthur A. Dugoni School of Dentistry, University of the Pacific, San Francisco, CA 94103, USA; kkonopka@pacific.edu; 2Institute of Biological, Environmental and Rural Sciences, Aberystwyth University, Aberystwyth SY23 3EE, UK; naf4@aber.ac.uk

**Keywords:** peptide design, virus entry, 6-helix bundle, coiled coil, computation methods, SARS-CoV-2, HIV-1, influenza

## Abstract

Fusion of lipid-enveloped viruses with the cellular plasma membrane or the endosome membrane is mediated by viral envelope proteins that undergo large conformational changes following binding to receptors. The HIV-1 fusion protein gp41 undergoes a transition into a “six-helix bundle” after binding of the surface protein gp120 to the CD4 receptor and a co-receptor. Synthetic peptides that mimic part of this structure interfere with the formation of the helix structure and inhibit membrane fusion. This approach also works with the S spike protein of SARS-CoV-2. Here we review the peptide inhibitors of membrane fusion involved in infection by influenza virus, HIV-1, MERS and SARS coronaviruses, hepatitis viruses, paramyxoviruses, flaviviruses, herpesviruses and filoviruses. We also describe recent computational methods used for the identification of peptide sequences that can interact strongly with protein interfaces, with special emphasis on SARS-CoV-2, using the PeP*I*-Covid19 database.

## 1. Cellular Entry of Lipid Enveloped Viruses

Virus replication can be generally divided into the early phase, which comprises attachment to the host cell, penetration and uncoating, and the late phase, which includes macromolecular synthesis, assembly and release [1,2]. Virus entry starts by attachment to particular receptors on the host cell membrane. The primary receptor for human immunodeficiency virus type 1 (HIV-1) is CD4, a marker of T helper lymphocytes that is also found on monocyte/macrophages. Glycoproteins and glycolipids with terminal sialic acid units act as host cell receptors for the influenza virus. Following receptor binding, enveloped viruses may fuse directly with the plasma membrane if their fusion machinery is active at neutral pH (e.g., HIV-1, measles virus), or they may be endocytosed and then fuse with the membrane of the endocytotic vesicle, if their fusion machinery requires the mild acidification of the endosome lumen facilitated by the cellular proton pump (e.g., influenza virus, rabies virus).

There are three main types of viral membrane fusion proteins [3,4,5]. The hemagglutinin, HA, of influenza virus, and the envelope protein (gp120/gp41) of HIV-1 are Class I proteins. They are synthesized as precursor proteins that are cleaved by cellular proteases, including the transmembrane protease serine S1 member 2 (TMPRSS2) [6], forming active trimers. In the case of HA, proper trimer formation is essential for transport from the endoplasmic reticulum to the Golgi and the plasma membrane [7]. The trimers are then incorporated into the budding viral membrane.

After interacting with their receptor, and in some cases after acidification of the local environment, these proteins fold into hairpin-like structures where two alpha-helices from each monomer interact with the two-alpha helices from the other two monomers, forming the hairpin conformation. This final structure is called the six-helix bundle. The transition to the six-helix bundle appears to be an important intermediate stage in membrane fusion and is thus a target for antiviral peptides.

The E1 protein of alphaviruses, which include Western, Eastern and Venezuelan equine encephalitis viruses and chikungunya virus, and the E protein of flaviviruses, which include West Nile, Zika, yellow fever, Dengue and hepatitis C viruses, are Class II fusion proteins that are produced when a precursor protein is cleaved [8,9,10,11]. The p62 is cleaved off the precursor polyprotein, p62-E1, to produce the E2 protein that protects the E1 protein and binds to receptors on the host cell membrane. Following endocytosis of the virions, these proteins change from an unstable dimer into a stable trimer that transforms into a hairpin-like structure; the fusion domain of the protein extends towards the endosome membrane, and the terminal hydrophobic fusion loop inserts into the membrane.

Class III glycoproteins are found in rhabdoviruses, such as rabies virus and vesicular stomatitis virus (the G protein), as well as herpesviruses (the gB protein), and do not require the cleavage of a precursor [5,12,13]. A stretched β-sheet fusion domain of class III proteins inserts hydrophobic fusion loops into the target membrane, similar to class II proteins. In rhabdoviruses, the G protein undergoes a conformational change that is initiated by the mildly acidic pH in endosomes.

In this review, we describe peptide inhibitors of membrane fusion of individual virus families, starting with HIV-1, for which a peptide medicine (“Enfuvirtide”) is already available for clinical use.

## 2. HIV-1

### 2.1. HIV-1 Infection Is Mediated by the Viral Envelope Glycoprotein (Env) That Attaches to the Cellular Receptor and Induces Virus–Cell Membrane Fusion

The Env protein of HIV-1 forms trimers in the viral membrane, and each protomer consists of two non-covalently interacting glycoproteins, the surface protein gp120 and the transmembrane protein gp41. Following the interaction of gp120 with the primary cellular receptor, CD4, Env undergoes conformational changes that enable the gp120 to bind to the co-receptor (CCR5 or CXCR4, depending on the tropism of the virus) and the insertion of the fusion peptide of gp41 to insert its fusion peptide into the host cell membrane, initiating the fusion reaction [14,15]. Single-molecule fluorescence resonance energy transfer experiments indicate that, after binding to a single CD4 molecule, the individual protomers have distinct conformations. The trimer undergoes further transformation when the remaining gp120 molecules bind CD4 and the co-receptor [15,16].

### 2.2. Small Peptides Can Inhibit Membrane Fusion

The carbobenzoxy-D-Phe-L-Phe-Gly peptide, an inhibitor of paramyxovirus fusion [17] was found to inhibit syncytium formation between chronically HIV-1-infected cells and CD4+ cells [18] (Table 1). The peptide, however, has a very limited effect on HIV-1 infection [18]. The latter observation suggests that the mechanisms of gp120/gp41-mediated cell–cell fusion may be different than that in virus–cell fusion [18]. By contrast, a hexapeptide with a sequence identical to the N terminus of gp41 inhibits the fusion of a pseudo-typed vaccinia virus expressing the HIV-1 envelope protein with HeLa cells expressing the cellular receptor, CD4 [19].

### 2.3. CD4- and V3 Loop-Derived Peptides Inhibit Membrane Fusion and Infection

Benzyl-derivatized peptides corresponding to residues 81–92 of CD4 inhibit HIV-1-induced cell–cell fusion and infection at micromolar concentrations [20]. A synthetic peptide corresponding to the CD4 complementarity-determining region 2 (CDR-2)-like domain binds specifically to HIV-1-infected cells [21]. When this peptide is coupled covalently to liposomes, it enables the binding of the liposomes to HIV-1-infected cells. A different peptide derived from the CD4 CDR-3-like domain inhibits HIV-induced syncytia formation at micromolar concentrations [21]. An eight-branched peptide derived from the V3 loop of gp120 completely inhibits HIV-1_LAI_ infection of a T lymphocyte cell line at 5 µM, while the monomer has a minimal effect, while a peptide corresponding to the entire V3 loop has no activity [22].

### 2.4. Peptides Corresponding to the Heptad Repeat Domain of Env Inhibit HIV-1 Membrane Fusion

Wild et al. [23] found that a peptide designated DP-178, corresponding to a possible alpha-helical region of the ectodomain of gp41, inhibits virus-induced cell–cell fusion with an IC_50_ of 100 pM or 0.5 ng/mL. Truncation of the C-terminal region of the peptide by four amino acids results in an inactive peptide. Heptad structures in proteins have a repeating amino acid sequence of the form (a-b-c-d-e-f-g)n, where “a” and “d” are apolar residues, and they form alpha-helices arranged in a coiled-coil structure [28]. The peptide DP-178 mimics the membrane proximal, C-terminal heptad repeat (CHR) region of gp41, and it most likely interacts with the membrane distal, N-terminal heptad repeat region (NHR) of gp41, in an intermediate conformation of gp41 before membrane fusion with the host cell membrane [24] (Figure 1).

If the inhibition of fusion is caused by interference with the interaction of the CHR and NHR domains, it may be expected that NHR-derived peptides should block the binding of CHR-peptides to the NHR domain. This inhibition, however, cannot be blocked by NHR-peptides [30]. Moreover, complexes of CHR-peptides and NHR-peptides are inhibitory to fusion. Thus, the mechanism of fusion inhibition may be more complicated than our current view. CHR-peptides also interact with the N-terminal, fusion peptide region of gp41. In a clinical trial, DP-178 (called “T-20” at this stage of its development as an antiviral) administered intravenously reduced the plasma levels of HIV-1 by almost two orders of magnitude [31]. In 2003, the U.S. Food and Drug Administration and the European Medicines Agency approved T-20 (Fuzeon; Enfuvirtide, San Francisco, CA, USA) for clinical use. 

The peptide segments N36 and C34 corresponding to the heptad repeat regions of the gp41 α-helical domain form a trimer of two interacting peptides, with three N36 helices forming an interior, coiled-coil trimer, and three C34 helices in an oblique, antiparallel fashion [32]. The six-helix bundle appears to form immediately after gp120 interacts with the co-receptor of HIV-1, CXCR4 [33].

### 2.5. Conjugation of Poly(ethylene glycol) to Heptad Repeat Peptides Prolongs Circulation in Blood

Since the half-life of T-20 in plasma is only about 1.1 h, Wang et al. [25] explored the effect of conjugating poly(ethylene glycol) (PEG) to peptide C34, which is analogous to a region of the CHR further towards the N-terminus compared to T-20. The half-life of the 2 kD PEG derivative of the peptide is 2.6 h, and that of the 5 kD PEG derivative is 5.1 h. The 50% effective concentrations (EC_50_) of these peptides in inhibiting HIV-1 Env-mediated cell–cell fusion are lower than those of T-20. 

### 2.6. Cholesterol Coupling to Heptad Repeat Peptides Can Enhance Their Fusion Inhibitory Activity

The C34 peptide competes with the CHR region of gp41, thus inhibiting the formation of the six-helix bundle involved in membrane fusion and comprising both the NHR and CHR [34]. Ingallinella et al. [26] attached a cholesterol molecule to the C-terminus of the C34 peptide in an attempt to localize the inhibitor to the cell membrane where HIV-1 fuses. C34-chol inhibits the infectivity of six HIV-1 isolates at IC_50_ values in the range 6–34 pM. These are much lower concentrations compared to those of plain C34 (205 pM) and T-20 (692 pM). Cholesterol coupling is, however, not a universal inhibition enhancer; for example, cholesterol-coupled T-20 has an IC_50_ of about 3726 pM.

### 2.7. A Peptide Targeting a Hydrophobic Pocket of HIV-1 gp41 N-terminal Heptad Repeat Trimers Inhibits Infection by Different Virus Isolates

The gp41 NHR-trimer has a hydrophobic pocket formed by each NHR’s 17-amino acid C-terminal region. The pocket has a conserved sequence, it is necessary for virus–cell membrane fusion, it constitutes a compact binding site for potential inhibitors and it is exposed to the medium, rendering it a good target for peptide inhibitors [35]. Welch et al. [27] designed a peptide, “PIE12-trimer”, composed of D-amino acids, with a very high affinity for the gp41 pocket. The IC_50_ of this peptide ranges from 0.4 nM for the NL4.3 isolate to 5.7 nM for the JRFL isolate, among the 23 viruses pseudo-typed with clonal and polyclonal envelopes, and is much lower than that of Enfuvirtide.

Thus, peptides corresponding to the C-terminal heptad repeat region of HIV-1 gp41 are effective inhibitors of viral infection.

## 3. Severe Acute Respiratory Syndrome Coronavirus (SARS-CoV)

### 3.1. The C-Terminal S2 Domain of the SARS-CoV Spike Protein S Is Involved in Membrane Fusion with the Host Cell Membrane

The coronavirus spike glycoprotein, S, is a Class I fusion protein and forms a trimeric structure visible in electron micrographs of the virus and changes its conformation after binding to the viral receptor, angiotensin-converting enzyme 2 (ACE2), on the cell membrane [29,36]. The transmembrane protease/serine subfamily (TMPRSS) cell surface proteases cleave the S protein into S1 and S2 sub-fragments. The receptor-binding domain (RBD) of S is located in the N-terminal S1 domain of S. Membrane fusion activity of the spike protein is associated with the C-terminal S2 domain (Figure 2). Insights on the mechanism of HIV-1 Env-mediated fusion led Liu et al. [37] to identify peptide inhibitors of the severe acute respiratory coronavirus (SARS-CoV) S-protein. The HIV-1 Env-mediated fusion is thought to involve the interaction of the membrane-distal, or N-terminal, NHR (also termed “HR1”) of gp41, and the membrane-proximal, or C-terminal, CHR (termed “HR2”) of the transmembrane protein. The S2 of SARS-CoV has some motifs similar to that of gp41, including the N-terminal leucine/isoleucine zipper and the C-terminal HR. Thus, it was likely that peptides like T-20 could potentially inhibit the fusion of SARS-CoV and the host cell membrane [38].

### 3.2. The HR2 Peptide Derived from the C-Terminal Heptad Repeat Region of S Inhibit SARS-CoV Infection

Looking at the cytopathic effect caused by SARS-CoV in Vero cells, Zhu et al. [39] found that a synthetic HR2 peptide and a glutathione-S-transferase (GST)-HR2 fusion peptide are inhibitory to infection by SARS-CoV (Table 2). The HR2 peptide has an EC_50_ in the range 0.5–5 nM, and the GST-HR2 peptide has an EC_50_ in the range 66–500 nM. In this system, there is no inhibitory effect for HR1-derived peptides. Using the pseudo-typed HIV-luc/SARS virus, Yuan et al. [40] reported an EC_50_ of 0.14 µM for the HR1-derived peptide, HR1-1, and 1.19 μM for the HR2-derived peptide, HR2-18. Experiments with wild-type SARS-CoV also showed that the peptides inhibit infection. 

### 3.3. The Peptides CP-1 and sHR2-8 Derived from the HR2 Region Inhibit Infection at Micromolar Concentrations

The 37-amino acid peptide (CP-1) corresponding to a sequence of the HR2 region of S inhibits infection at micromolar concentrations [37]. Biophysical techniques including surface plasmon resonance and sedimentation equilibration analysis showed that this peptide binds to a peptide (NP-1) derived from the HR1 domain with high affinity and forms a six-helix bundle when mixed at equimolar concentrations. During the structural rearrangement of the S protein as SARS-CoV is fusing with the cell membrane, CP-1 most likely binds to the HR1 region and prevents the association of the HR2 and HR1 heptad repeat domains, and thus inhibits membrane fusion. A study by Bosch et al. [41] identified an HR2-derived peptide (sHR2-8) with an EC_50_ of 17 µM, based on SARS-CoV infection of Vero cells.

### 3.4. The HR2-Derived Peptides SR9 and SR9EK13 Inhibit Viral Entry at the Cell Surface, following Protease Activation of S at Nanomolar Concentrations

A 23-mer peptide (P6), derived from the HR2 region, inhibits membrane fusion between S-protein-expressing cells and cells expressing ACE2 at an IC_50_ of 1 μM [42]. The peptide most likely inhibits fusion by binding to the deep groove of the HR1 trimer. The combined use of an HR1-derived peptide, N46, and its mutant N46eg, inhibits cell–cell fusion at an IC_50_ of 1.4 μM, in contrast to previous studies. Ujike et al. [43] showed that two HR2-derived peptides, SR9 and SR9EK13, inhibit the plasma membrane entry of SARS-CoV with an EC_50_ of 4–5 nM, following the activation of the viral S protein by proteases. However, they are ineffective against the endosomal entry of the virus. These peptides may be effective in preventing viral infection of the lungs after activation of the spike protein by lung proteases [43].

It is clear that peptides based on the HR2 domain of the cleaved S protein (S2) of SARS-CoV have significant antiviral activity. It is not known why some peptides have effective concentrations in the micromolar range and others in the nanomolar range.

## 4. Middle East Respiratory Syndrome Coronavirus (MERS-CoV)

### 4.1. The Peptides HR1P and HR2P Derived from the HR1 and HR2 Regions, Respectively, of the MERS-CoV S2 Protein Form a Six-Helix Bundle, and HR2P Inhibits Membrane Fusion

Middle East respiratory syndrome coronavirus (MERS-CoV) infects cells through the interaction of its S protein with the cellular receptor, dipeptidyl peptidase 4 (CD26) [44,45], cleavage of the protein by cell membrane proteases and membrane fusion between the virus and the host cell. X-ray diffraction of the crystallized S2 subunit of the S protein indicated that the peptides HR1P derived from the HR1 region of the S2 protein and HR2P derived from the HR2 region form a six-helix bundle [46]. Virus replication and cell–cell fusion are inhibited by HR2P (Table 3). The cell–cell fusion inhibitory activity of the peptide is enhanced by adding hydrophilic amino acids to the peptide to increase its alpha-helicity, with an IC_50_ of 0.56 µM. Another derivative, HR2P-M2, with improved pharmaceutical characteristics, inhibits S-protein-mediated cell–cell fusion (IC_50_ = 0.55 µM) and infection by a pseudo-virus that expresses S (IC_50_ = 0.6 µM) [47]. After intranasal delivery, the peptide reduces the viral titer in the lungs in a murine model of MERS-CoV infection by more than three orders of magnitude.

### 4.2. The MERS-Five-Helix Bundle, with Three HR1 and Two HR2 Peptides, Inhibits Infection by a Pseudo-Typed Virus

Sun et al. [48] designed an unusual peptide conglomerate termed “MERS-five-helix bundle (MERS-5HB)”, comprising three copies of HR1 and two copies of HR2, thus lacking one HR2, which would have been part of the six-helix bundle. Thus, MERS-5HB could be expected to bind the viral HR2, and thus inhibits the formation of the six-helix bundle. MERS-5HB interacts with an HR2 domain peptide, HR2P, with a K_D_ of about 0.24 nM, and inhibits infection by pseudo-typed MERS-CoV at an IC_50_ of about 1 μM, and cell–cell fusion at an IC_50_ of approximately 0.6 µM.

The concept of inhibiting the formation of the six-helix bundle to inhibit viral membrane fusion holds for MERS-CoV as well.

## 5. Severe Acute Respiratory Syndrome Coronavirus-2 (SARS-CoV-2)

### 5.1. The Cholesterol-Coupled, HR1-Targeting Peptides EK1C4 and IPB02 Inhibit Cell–Cell Fusion and Pseudo-Typed Virus Fusion at Nanomolar Concentrations

Like SARS-CoV, SARS-CoV-2 binds the ACE2 receptor via the S1 domain of its spike protein, S [51]. SARS-CoV-2 appears to fuse more efficiently with the host plasma membrane than SARS-CoV [49]. The peptide EK1 designed by Xia et al. [49] targets the HR1 domain of the S-protein and inhibits the fusion activity of SARS-CoV and MERS-CoV (Figure 3, Table 4). The cholesterol derivative of EK1 (EK1C4) is a highly effective inhibitor of S-protein-mediated cell–cell fusion, with an IC_50_ of 1.3 nM, which is 241-fold lower than that of the original peptide, EK1. Infection by a pseudo-virus expressing S is inhibited by EK1C4 at an IC_50_ of 15.8 nM, 149-fold lower than that of EK1.

The HR1 domain of SARS-CoV-2 has higher α-helicity and binding affinity to the HR2 domain than the HR1 region of SARS-CoV [51]. The lipopeptide, IPB02, derived from the HR2 sequence and modified by coupling cholesterol to its C-terminus, is highly effective in inhibiting SARS-CoV-2 S-mediated cell–cell fusion (IC_50_ = 25 nM) and infection by an S-expressing pseudo-virus (EC_50_ = 80 nM). 

### 5.2. There Are Assay- or Cell-Dependent Variations in the Evaluation of Membrane Fusion Activity

A cholesterol-coupled lipopeptide corresponding to the C-terminal heptad repeat (HRC or HR2) domain of SARS-CoV-2 S inhibits S-mediated cell–cell fusion, measured by β-galactosidase complementation, with an IC_50_ of ∼10 nM and an IC_90_ of ∼100 nM [50]. Cholesterol-coupled EK1 [49] also inhibits cell–cell fusion in this system [50], but at an IC_50_ of ~300 nM (IC_90_ > 900 nM), indicating assay- or cell-dependent variations in measurements of membrane fusion activity. An advantage of this HRC-lipopeptide is that it is highly inhibitory to fusion by the D614G, S943P and S247R variants of SARS-CoV-2, as well as MERS and SARS-CoV S-mediated cell–cell fusion. The HRC-lipopeptide is highly effective in inhibiting SARS-CoV-2 infection in a plaque neutralization assay, with an IC_50_ of ~6 nM [50]. It also inhibits viral spread in human airway epithelial cultures (which is an ex vivo model of virus propagation in the lungs), measured by using SARS-CoV-2 expressing a stable mNeonGreen reporter gene and monitoring the fluorescence of the cultures.

The low nanomolar inhibitory concentrations of these molecules, especially those containing cholesterol that facilitates membrane localization, render them candidates for antiviral drugs.

## 6. Influenza Virus

### 6.1. Influenza Virus Fusion Is Mediated by the Low pH-Induced Conformational Change of the Viral Hemagglutinin

The hemagglutinin (HA) of influenza virus is cleaved via proteolysis by cellular enzymes into HA1 and HA2 subunits. These enzymes include TMPRSS2, TMPRSS4, the human airway trypsin-like protease and kallikrein-related peptidases 5 and 12 [52,53]. HA1 mediates binding to sialic acid-containing glycoproteins or glycolipids in the host membrane. The mildly acidic pH encountered in endosomes following endocytosis of the virus causes the unfolding of the trimeric HA [54,55] and the insertion of the hydrophobic N-terminus of the HA2 (the fusion peptide) into the host cell membrane. We have proposed that it is the conformational change of the HA that mediates fusion, and not the low pH configuration itself [56,57]. Supporting this hypothesis is the finding that if influenza virus is exposed to low pH without the presence of the target membrane, the HA undergoes an irreversible conformational change and the virus is unable to fuse [58,59].

### 6.2. Influenza Virus Fusion May Be Inhibited by Competitive Inhibitors of the Receptor-Binding Domain of HA

Matsubara et al. [60] utilized a phage display peptide library to identify peptides that fulfilled this function (Table 5). The peptide s2 coupled to stearic acid (C18-s2) inhibits the infection of MDCK cells by the H1N1 influenza virus at an IC_50_ of 11 µM, and by the H3N2 virus at 15 µM. EB (“entry blocker”), a 20-amino-acid peptide, inhibits virus binding to the cellular receptor by interacting with HA [61]. The peptide inhibits virus-induced cytotoxicity to MDCK cells with an IC_50_ of 4.5 μM, and is effective against infection by H5N1, H5N9 and H1N1 subtypes.

Based on molecular dynamics and quantum chemistry calculations, Perrier et al. [62] designed peptides that could bind a protein cavity in the HA2 subunit of HA. They identified two pentapeptides that interacted with HA2 with high stability at both pH 7 and 5; however, the peptides have not been tested for their fusion inhibitory activity.

### 6.3. A Cholesterol-Coupled Peptide Corresponding to a Membrane Proximal Domain of HA Inhibits Influenza Infection at Low Micromolar Concentrations

The internal coiled coil of HA identified in the low pH-induced, post-fusion structure of the protein interacts with a region of HA immediately outside the viral membrane [63]. The peptide (P155-185) corresponding to a segment of this region, coupled to cholesterol at its C-terminus (generating P155-181-chol), inhibits infection by influenza A/H3N2 with an IC_50_ of 0.4 μM. A peptide missing the four C-terminal amino acids has an IC_50_ of 2 µM. The peptides without the cholesterol do not have a significant inhibitory effect. The authors attributed this effect to the insertion of the cholesterol moiety into the plasma membrane and internalization of the coupled peptide together with the virus in endosomes. The acidification of the endosome lumen exposes the region of HA recognized by the peptide, thereby preventing the full conformational change of the protein that would otherwise induce membrane fusion [63].

### 6.4. A Fusion Inhibitory Peptide Derived from a Helical Region of HA2 Coupled to a Cell-Penetrating Peptide Reduces the Viral Titer in an Animal Model

Figueira et al. [64] added a cell-penetrating peptide to a fusion inhibitory peptide of 43 amino acids, which was derived from a helical region of the HA2 subunit participating in helix–helix interactions. The cell-penetrating peptide, which would facilitate partition of the entire peptide into the plasma membrane, was derived from the HIV-1 Tat protein. The new construct inhibits influenza virus fusion with liposomes composed of dioleoylphosphatidylcholine, a zwitterionic lipid, at a relatively high concentration (10 µM), which may be expected because of the absence of a sialic acid-containing receptor on the liposome surface. The intranasal administration of the peptide, however, reduces significantly the local viral titer in infected rats.

### 6.5. The Peptide iHA-100 Generated by a Macrocyclic Peptide Expression System Inhibits Membrane Fusion at Nanomolar Concentrations

The Random Non-Standard Peptides Integrated Discovery (RaPID) system is a peptide screening system that combines mRNA display technology with the Flexible In Vitro Translation System [65]. This system consists of “flexizyme”, an artificial ribozyme and a cell-free translation system from *Escherichia coli*, and enables the expression of thioether-macrocyclic peptides including non-standard amino acids [65]. This system helps in the identification of high-affinity ligands for a protein of interest. Using the RaPID system, Saito et al. [66] obtained macrocycles named iHAs that can bind influenza HA. Among these peptides, iHA-100 inhibits both viral adsorption (IC_50_ = 0.036 µM) and membrane fusion (EC_50_ = 0.07 µM) by interacting with the HA stalk domain. 

### 6.6. The Complementarity Determining Domains of Neutralizing Antibodies Can Be Used to Design Peptides

In a different approach to peptide design, Kadam et al. [67] utilized the complementarity-determining region loops of human broadly neutralizing antibodies against influenza HA. The optimized peptides, at nanomolar concentrations, bind to the stem region of HA of 2009 H1N1 pandemic and avian H5N1 strains and block the low pH-induced conformational change of HA that leads to membrane fusion. 

## 7. Hepatitis Viruses

### 7.1. The Region of the Hepatitis Virus L Protein That Binds to the Cell Surface Sodium-Taurocholate Co-Transporting Polypeptide Receptor has been Identified

The Hepatitis B virus (HBV) membrane has large (L), middle (M) and small (S) membrane-spanning glycoproteins that share the same C-terminal S-domain, which is anchored in the lipid bilayer by five membrane-spanning helices. The L protein contains the N-terminal pre-S1, the central pre-S2 and the C-terminal S domains. M lacks the pre-S1 domain, and S only has the S domain [68]. These proteins are glycosylated type II transmembrane proteins. They form multimers with disulfide bridges. The virus interacts first with heparan sulfate proteoglycans on the hepatocyte surface. The myristoylated N-terminal of the pre-S1 domain [69,70] binds to the high-affinity receptor, the sodium-taurocholate co-transporting polypeptide (NTCP/SLC10A1) [70,71]. The identification of NTCP as a hepatocyte receptor has revealed a target for HBV entry inhibition [68,72,73,74,75].

### 7.2. Myrcludex B Inhibits the Binding of Hepatitis B Virus to Its Receptor

The myristoylated peptide, Myrcludex B, whose sequence corresponds to the N-terminal amino acids of the pre-S1 of L, inhibits HBV and hepatitis D virus entry into hepatocytes by binding with high efficacy to NTCP (IC_50_ = 80 pM) [74,76] (Table 6). Myrcludex B (currently named bulevirtide) is currently in phase III clinical trials in chronically HBV-/HDV-infected patients [77].

### 7.3. The E2 Protein of Hepatitis C Virus Interacts with Different Cellular Receptor Molecules, including Tetraspanin, Claudin and Transferrin Receptor 1

Hepatitis C virus (HCV), a small, enveloped virus containing a positive-sense, single-stranded RNA, is classified in the *Hepacivirus* genus of the *Flaviviridae* family. The envelope glycoproteins E1 and E2 are type I transmembrane proteins with an N-terminal ectodomain and a short C-terminal transmembrane domain. They form large covalent heterodimers stabilized by disulfide bridges [80]. The entry of HCV into hepatocytes requires interactions between the viral envelope proteins and several host cell surface molecules, including tetraspanin CD81, scavenger receptor class B type 1 (SR-B1), claudin-1 and occludin, transferrin receptor 1 (TfR1), receptor tyrosine kinases (RTKs), Niemann-Pick C1-like 1 (NPC1L1) cholesterol uptake receptor and epidermal growth factor receptor [68,69]. E2 is the major HCV envelope protein that directly interacts with these receptor molecules. HCV associates with lipoproteins, forming hybrid particles called lipo-viro-particles (LVP). The first step of HCV entry involves the interaction between LVP-associated ApoE, cell surface heparan sulfate proteoglycans (particularly syndecans 1 and 4) and the LDL receptor. LVP then interacts with SR-B1 via ApoE and ApoB-100. The cholesterol transfer activity of SR-B1 enables E2 exposure and binding to SR-B1 and CD81. Binding to CD81 activates the epidermal growth factor receptor signaling pathway, and the interaction between CD81 and claudin-1 mediates HCV endocytosis. This is followed by low pH-induced fusion between the viral and endosome membranes [78,81,82,83,84,85,86,87,88,89]. 

### 7.4. Peptides from the E2 Protein and from Claudin-1 Inhibit Infection at Micromolar Concentrations

Chi et al. [90] identified a peptide from the E2 stem domain named E27 that blocks E1–E2-mediated cell–cell fusion, and inhibits entry of HCV pseudo-particles, as well as infection by cell culture-derived HCV, with an EC_50_ of 0.73 µM. E27 likely prevents fusion by interfering with E1–E2 dimerization and by inducing conformational changes of the E1–E2 dimer. Better activity was achieved by shortening this peptide from 35 to 29 amino acids. An 18-amino acid peptide derived from the N-terminus of claudin-1 inhibits a late step during viral entry, most likely after the binding step, with an IC_50_ of 2.1 µM [79]. The peptide, designated as CL58, is not cytotoxic at concentrations about 100-fold higher than the IC_50_. Slightly shorter peptides have much higher IC_50_s, and slightly longer peptides have slightly higher IC_50_s, highlighting the importance of peptide length in the design of inhibitors.

The lipopetide Myrcludex B inhibits hepatitis B virus entry, and the peptides CL58 and E27 inhibit hepatitis C virus fusion.

## 8. Paramyxovirusess

### 8.1. The Envelope Protein H Mediates Virion Attachment to the Host Cell Membrane, and the F Protein Carries out the Membrane Fusion Reaction That Involves the Anti-Parallel Alignment of Heptad Repeat Domains

Paramyxoviruses are lipid-enveloped, negative-strand RNA viruses in which the single strand of RNA forms a helical nucleocapsid with the nucleoproteins and is transcribed into positive-strand mRNA by the RNA-dependent RNA polymerase [1,2]. They include the medically important viruses, measles, mumps and respiratory syncytial viruses. The lipid envelope is pleomorphic and incorporates hemagglutinin, H, neuraminidase and the fusion protein, F. The virus also has the nucleoprotein, large protein, P protein and the matrix protein [91]. Paramyxoviruses bind to cell surface receptors including sialic acid, CD46 and CD150. Mumps virus uses α2,3-linked sialic acids in oligosaccharides on lipids or glycoproteins. Clinical isolates of measles virus bind CD150 and the adherens-junction protein, nectin-4, whereas laboratory and vaccine strains use CD46 as their receptors [92].

The H protein mediates attachment to the host cell, and the F protein initiates the membrane fusion reaction after the cleavage of a precursor that exposes the fusion peptide, which is inserted into the host cell membrane [57,93,94], providing an intermediate structure that pulls together the viral and cellular membranes. The two heptad-repeat domains on the F protein fold onto each other and onto the neighboring F proteins in an anti-parallel fashion. The resulting six-helix bundle brings the membranes closer. The formation of this structure can be inhibited by complementary peptides [95], as in the cases of the other viruses described above. 

### 8.2. A Short Peptide Analogous to the N-Terminus of the Cleaved F Protein Can Inhibit Sendai Virus Fusion

The earliest discovery that a peptide could inhibit viral infection was made by Richardson et al. [17]. They found that a peptide with an amino acid sequence similar to that at the N-terminus of the paramyxovirus F1 (carbobenzoxy-D-Phe-L-Phe-Gly) inhibits viral replication. It also inhibits the fusion of the paramyxovirus, Sendai virus, with liposomes [96]. The inhibitory mechanism has been correlated with the inhibition of the bilayer-to-hexagonal II phase transition of a species of phosphatidylethanolamine [97]. Whether this potential mechanism of inhibition is relevant to actual viral membrane fusion is not known. 

Rapaport et al. [98] found that a peptide corresponding to the Sendai virus F protein heptad repeat closest to the viral membrane inhibits its fusion with erythrocytes. Peptides derived from the paramyxovirus F1 protein, with amino acid sequences similar to that of peptide inhibitors of HIV-1 (DP-107 and DP-178 (T-20)), have antiviral activity, blocking syncytium formation by respiratory syncytial virus, human parainfluenza virus type 3 and measles virus, with EC_50_s between 15 and 250 nM [99]. Synthetic peptides of 34 amino acids corresponding to the heptad repeat domains of the F proteins of human parainfluenza virus type 2 and type 3 proximal to the transmembrane segment of the protein inhibit syncytium formation by these viruses at EC_50_ of 2.1 µM and 1.2 µM, respectively, and inhibit viral entry [100].

### 8.3. The Peptides HRC4 and HRC2 Derived from the Measles Virus F-Protein C-Terminal Heptad Repeat Region Inhibit Measles Virus Infection at Nanomolar Concentrations

Dimer peptides corresponding to the measles virus F-protein C-terminal heptad repeat region inhibit measles virus infection at IC_50_ values in the range 1–2 nM (for peptides HRC4 and HRC2) when coupled to cholesterol that presumably enables insertion of the conjugate into the cell plasma membrane. HRC4-cholesterol prevents measles virus-induced death in a murine model when administered intranasally at a dose of 6 mg/kg, 24 h before infection [101]. When peptides are coupled to lipids, they can self-assemble into micellar nanoparticles and can partition into cell membranes [102]. Membrane-localized peptides can interact more effectively with the viral heptad repeat regions compared to plain peptides [103]. Cholesterol-derivatized HRC peptides inhibit the entry of parainfluenza virus type 3 into CV1 cells at an IC_50_ of about 7 nM, whereas plain peptides have a 100-fold higher IC_50_ [104]. The highest antiviral activity is achieved when cholesterol is attached to the C-terminus of the peptide, suggesting that the positively charged amino terminus may be involved in the interaction of the peptide with its target in the F protein, or that the anti-parallel orientation of the peptide helix is important for inhibition.

## 9. Flaviviruses

### 9.1. The Class II E Protein of Flaviviruses Binds to Phosphatidylserine Receptors on Host Cells, and its Low pH-Induced Conformational Change in Endosomes Mediates the Interaction between Two Domains of the Protein

Dengue, yellow fever and West Nile viruses are flaviviruses that are enveloped, positive-strand RNA viruses that infect their host cells via receptor-mediated endocytosis. The E protein of the viral envelope binds to attachment factors, including glycosaminoglycans on the cell membrane [105]. The TIM and TAM proteins are families of receptors that are involved in the phagocytotic removal of phosphatidylserine-expressing apoptotic cells. These proteins act as receptors for the phosphatidylserine present in the Dengue virus membrane [106,107]. This negatively charged lipid is most likely acquired during viral biosynthesis in the endoplasmic reticulum [108]. The E protein, a class II fusion protein, undergoes a low pH-induced conformational change that mediates the interaction between the stem region and domain II; this change also mediates a transition from dimers to trimers [109,110,111] that initiates the membrane fusion reaction between the virus and the endosome membrane.

### 9.2. Peptides Derived from the E Protein Stem Helix 2 Inhibit Infection of Japanese Encephalitis Virus at Nanomolar IC_50_

Costin et al. [112] designed peptides that interact with the E protein of Dengue virus and inhibit viral infection in culture (Table 7). They used high-resolution structural information for the dimeric form of the E protein, and computational optimization of binding. The peptides DN57opt and 1OAN1 have IC_50_s of 8 µM and 7 µM, respectively, inhibit virus binding to cells, and bind directly to the E protein. Chen et al. [111] investigated the ability of peptides derived from the E protein stem helix 2 of Japanese encephalitis virus to inhibit infection. These peptides inhibit infection at nanomolar IC_50_s. The peptide P5 prevents tissue pathology, decreases the viral load in an animal model, and inhibits lethality. P5 also inhibits Zika virus infection at micromolar IC_50_ and prevents histopathological damage in the brains of infected mice.

## 10. Herpesviruses

### 10.1. Herpesviruses Utilize a Complex Set of Membrane Proteins to Eventually Activate the gB Protein, which Mediates Membrane Fusion by Inserting into the Host Cell Membrane and Changing Conformation

Herpesviruses are enveloped, double-stranded DNA viruses with an icosahedral nucleocapsid [2]. They bind to specific receptors on the surface of host cells, including heparan sulfate and the intercellular adhesion molecule, nectin-1. The heterodimeric viral membrane proteins gH–gL and the protein gB that mediates membrane fusion are required for the cellular entry of the different herpesviruses [113,114]. The tropism of these viruses, however, is determined by specific sets of viral glycoproteins, whose conformational changes upon binding to cellular receptors appear to activate the Class III fusion protein gB [115] to initiate membrane fusion. In the case of herpes simplex virus type 1 (HSV-1), the gD binds to cell surface receptors, and then interacts with the gH–gL heterodimer, which in turn activates gB [116]. A segment of gB inserts into the host cell membrane, and folds back on itself to cause the viral and cellular membranes to come together. Several folded gB trimers result in the formation of a fusion pore [113]. Epstein–Barr virus binding to B cells is mediated by the glycoprotein gp42, which interacts with gH–gL, and the interaction of the resulting trimeric complex with the receptor triggers gB to effect fusion. For entry into epithelial cells, however, Epstein–Barr virus utilizes gH–gL to bind directly to the receptor, thereby activating gB [113].

### 10.2. Peptides It1b and MelN4 Designed to Have Membrane “Interfacial Activity” Inhibit HSV-1 Infection at Low Micromolar Concentrations

The glycoproteins gB and gH of HSV-1 have heptad repeat sequences predicted to form α-helical coiled coils [117,118]. Peptides corresponding to the coiled-coil domain of gH and gB inhibit viral infection, but at very high doses (e.g., 65% inhibition at 500 µM) [119]. Peptides analogous to the hydrophobic, membrane-interactive segments of gH (“fusion peptides”) inhibit viral plaque formation, with an IC_50_ of 160 µM for gH493-512 and 60 µM for gH626–644. It was suggested that the inhibitory effect may be the result of peptide association with the “fusion peptide“ of gH [120]. The rather high inhibitory concentrations of these peptides may be attributable to the subsequent observations that it is the gB protein that mediates membrane fusion. Certain peptides designed de novo for having interfacial activity (i.e., interacting with the interfacial region of lipid bilayers) can inhibit HSV-1 infectivity; for example, the peptide It1b has an IC_50_ of <2 µM, and MelN4 has an IC_50_ of ~3 µM [121] (Table 8).

### 10.3. The Peptide gBh1m Corresponding to the N-terminal Domain of the Coiled-Coil Structure of gB and Modified with Poly(ethylene glycol) and Cholesterol Inhibits HSV-1 Infection

The Galdiero laboratory started with a peptide (gBh1m) based on the N-terminal region of the coiled coil that may be able to arrest gB in its pre-fusogenic state [119], and modified it by conjugating 12-mer or 24-mer poly(ethylene glycol) and cholesterol to either end of the peptide [115]. The peptide gBh1m-Cys-PEG24-Chol modified at the C-terminus appeared to be the most effective among the reagents with different molecular arrangements, having an IC_50_ of approximately 8 µM, when the peptide was added to the culture at the time of infection. This peptide also had the highest “selectivity index”, defined as the ratio of the 50% cytotoxic concentration (CC_50_) to the IC_50_.

The relatively high concentrations of the peptides designed against herpesvirus fusion compared to those for HIV-1 and coronaviruses is most likely related to the rather complex structure of the Class III fusion protein of herpesviruses [113,114,122].

## 11. Filoviruses

### 11.1. The Ebola Virus Membrane Protein GP1 Is Cleaved in Endosomes and the Cleaved Form GPcl Binds the Cholesterol Transporter Niemann-Pick C1

Filoviruses are filamentous, enveloped, negative-strand RNA viruses, and include Marburg and Ebola viruses [2]. Ebola virus attaches to the TIM and TAM family of phosphatidylserine receptors that interact with this lipid in the viral membrane, and lectins that bind the viral glycoprotein GP [123,124]. The latter comprises GP1, the receptor-binding subunit, and GP2, the fusion subunit. Following internalization of the virus by macropinocytosis, and possibly by clathrin- or caveolin-mediated endocytosis [123,125], GP1 is cleaved by cathepsins B and L in endosomes, and the cleaved form (GPcl) binds the endosomal cholesterol transporter protein, Niemann-Pick C1. This interaction, however, does not appear to be sufficient to induce membrane fusion [124]. In a cell–cell fusion model, mildly acidic pH was found not to be the trigger for membrane fusion [126].

### 11.2. Conjugation of the HIV-1 Tat Peptide to an Ebola Virus CHR Peptide Mediates Endosome Localization and Enhances Antiviral Activity

Fusion of the Ebola virus with cellular membranes is likely to be facilitated by conformational changes of GP2, resulting in the six-helix bundle structure involving the heptad repeats near the N-terminus (NHR) and that near the C-terminus (CHR) of the protein. Since C-peptides corresponding to the CHR have very low antiviral activity, Miller et al. [127] attached the peptide to the arginine-rich (hence, positively charged) sequence from the HIV-1 Tat protein. This conjugate (Tat-Ebo) specifically inhibited viral entry mediated by the GP protein (with an IC_50_ of about 30 µM) and Ebola virus infection (with an IC_50_ of about 20 µM), possibly resulting from the accumulation of the peptide conjugate in endosomes (Table 9).

### 11.3. Peptides Can Have Inhibitory Effects across Virus Families

A peptide (RVFV-6) based on the stem region of the Rift Valley fever virus fusion glycoprotein, Gc, inhibits not only infection by this virus, but also Ebola virus infection in Vero E6 cells, with an IC_50_ of 11 µM [128]. The peptide is thought to interact with both the viral and cellular membranes, and then binds specifically to the Gc that undergoes a conformational change at low pH in the endosomes. C-peptides conjugated to cholesterol inhibit Ebola virus glycoprotein-mediated infection of pseudo-typed vesicular stomatitis virus, with an IC_50_ of about 6 µM, and reduce infection by three orders of magnitude at 40 µM. The observation that these peptides also inhibit fusion by vesicular stomatitis virus glycoprotein G protein indicate that the action of the peptides is not specific to Ebola virus fusion.

### 11.4. Cyclic Peptides That Bind GPcl Can Inhibit Infection by a Pseudotyped Virus

Li et al. [129] used docking and molecular dynamic simulations to design cyclic peptides that could bind the cleaved Ebola virus protein GPcl at the site that interacts with the Niemann-Pick C1 protein, based on the crystal structure of the two interacting proteins. The most effective of these peptides is Pep-3.3, which inhibits infection by GP-pseudo-typed HIV-1 at an IC_50_ of 5.1 μM.

Despite the very high mortality rate of Ebola virus infection, it is disappointing that the experimental peptides have not progressed to clinical trials, or emergency use.

## 12. Protein Design Methods to Target Protein Interfaces

Peptide-mediated interactions between proteins is a common mechanism, accounting for up to 40% of the protein interactions in cells [130]. More broadly and in the context of protein–protein interfaces, peptides represent an important tool to modulate interactions as there is a growing perception that traditional low molecular weight, drug-like chemical compounds are not well suited to target protein interfaces, given their size [131]. Peptides present a number of advantages that make them the more natural candidates to target protein interfaces [132], with over 60 peptide therapeutics on the market, and several hundred in preclinical and clinical development [133]. 

Computational approaches can aid in the discovery and design of peptide ligands to target protein interfaces. These will be briefly presented in this section, together with a more detailed description of a method published earlier [134]. This method was applied to the S spike protein of SARS-CoV-2 to derive a repository of peptides to target the interaction with the human angiotensin-converting enzyme 2 receptor. The repository is termed the Pep*I*-Covid19 database (http://bioinsilico.org/pepicovid19, accessed on 1 December 2021) [135]. There are two basic approaches to design peptides, depending on the information required:

(1) Methods based on the primary structure of proteins, i.e., the sequences; and

(2) Those based on the three-dimensional structure of the proteins.

## 13. Sequence-Based Methods

### 13.1. Relying on the Primary Sequence of Proteins, Synthesized Peptides Can Mimic the Key Contacts between Interacting Proteins and Inhibit Their Normal Functions

These methods rely on the primary sequence of proteins to design peptides. A common approach relies on the mimicry of one of the partners in a given protein–protein interaction. The idea behind this is rather simple and involves the identification of short stretches of residues on either of the interacting protein partners that, once synthesized as peptides, will act as a surrogate by mimicking the key contacts between partners, and therefore impinging upon the normal interaction of the full proteins. This approach presents a number of limitations where perhaps the most important is to assume that the isolated peptide will retain the same conformation as when it is part of the protein. Additionally, given the fact that the only information used is the sequence, the uncertainty with regards to the context and specific location of the putative interacting peptides is very high. Nonetheless, this method has been pursued and used successfully to identify peptides to modulate, among others, the interactions between protein kinases and their substrates [136,137], the inhibition of beta-APP-cleaving enzyme-1, an important target in Alzheimer’s disease [138], the inhibition of G-protein-coupled receptors [139], the inhibition of integrins using derivatives and constrained (cyclic) peptides presenting the conserved Arg-Gly-Asp motif or the interactions between viral interleukin-6 and the gp130 receptor [140], among others [141] (see the review by Rubinstein and Niv [142] for more details). Examples were presented earlier in this review where the peptides were derived from particular regions of the proteins, like Enfuvirtide derived from the CHR domain of gp41, which inhibits the fusion of HIV and the host cell plasma membrane [23,24,143]. Using the sequence method, we designed a peptide that competes with and disrupts the aggregation of β-amyloids [144]. 

### 13.2. Large-Scale Peptide Screening, Utilizing Phage-Display, mRNA or DNA-Encoded Libraries, Can Generate Inhibitory Molecules following Large-Scale Screening

The discovery of inhibitory peptides can also be approached using large-scale peptide screening. This approach is slightly different and relies on the high-throughput synthesis of peptide libraries that are subsequently used to probe the protein of interest. Peptide libraries can be constructed using phage-display [145], mRNA [146] or DNA-encoded libraries [147], bead-based libraries [148,149] or peptide arrays [150]. The sequences sampled by these libraries can be totally random or biased toward a specific size and/or amino acid composition, depending on the information available. Their diversity ranges from 10^8^ to 10^13^ combinations; so, the downstream processing of screening is important to identify binders through affinity selection and mass spectrometry [151]. Indeed, the necessity of a scalable downstream assay to assess the bioactivity of the peptides is a central and, often, the most difficult and challenging aspect of designing therapeutic peptides. The size of the peptides in these libraries is often predetermined and can result in the discovery of not only the recapitulation of variants of native ligands, but also, and more importantly, the discovery of novel sequences that are not similar to the native ligands and/or interface regions. Examples of large-scale screening identifying protein–protein inhibitors include the targeting of the interaction between p53-MDM2 and MDMX complexes [151], 14-3-3-mediated interactions [152] and protein kinase B-Akt interactions [153]. Finally, the advent of “omics” technologies, and more precisely genomics, transcriptomics and peptidomics, has allowed for the discovery of novel peptide sequences to modulate protein interactions [154,155]. Instead of relying on the discovery of peptides based on de novo synthesis, these approaches identify potential peptide ligands from the mining of sequences from natural repertoires. While these methods do not require the generation of libraries of variants, they share with the de novo discovery methods the same challenges of high-throughput synthesis and downstream screening.

Clearly, sequence-based identification of peptides is a very useful method in generating inhibitory molecules to a first approximation.

## 14. Structure-Based Methods

### 14.1. The Precise Structural Information of the Target Protein or Protein Complex Can Be Used in Modeling and Designing Inhibitory Peptides

The structure of proteins and/or protein complexes provides extremely valuable information for the design of peptides. On the one hand, the structure of a protein complex defines very precisely the structural features of the protein interface(s). On the other hand, it allows for the identification of the structural determinants and dominant secondary structure elements, e.g., α-helix, driving the interactions between partners. This information is then included in the modeling and designing of the peptides. The main positive aspect of these methods is, therefore, the precise structural information of the target protein or protein complex that can be fed into the modeling and designing process of the peptides. The negative aspect is, of course, the fact that obtaining the three-dimensional structure of proteins or protein complexes is far from trivial; thus, the range of applicability of these methods is restricted. 

### 14.2. Chemical Modifications of the Peptides, including Carbon-Stapling and Peptide Cyclization, Can Preserve Their Structural Integrity

Analogous to the sequence-based method that relies on the use of residue stretches to identify decoys of the protein interfaces, the structure of the protein complex enables the precise designation of the regions of the protein sequence that can play an important role in the interaction. Once the region of interest is identified, the peptide can be synthesized, and the inhibitory properties assessed. More important, however, as the structure of the fragment is known, it is possible to design chemical modifications, including carbon-stapling [156] or peptide cyclization [157], to ensure the structural integrity of isolated peptides, as it is not guaranteed that the conformation of the fragment will be preserved once isolated from the protein, i.e., as a peptide. Some examples are the stapling of the α-helix central to the interaction between NoxA and MCL-1 [158], and the cyclic peptide derived from a loop region mediating the interaction between the human 14-4-3-zeta protein and the virulence factor exoenzyme S of *Pseudomonas aeruginosa* [159]. Finally, the structure of the complex also facilitates the identification of the regions that contribute the most to the interactions, either at the residue level (so-called *hot spots*) [160] or stretches [161], with information that can be used to guide the selection of putative binders.

In terms of targeting protein interfaces using structure-based-designed peptides, there is a wide range of methodologies. Among those are the de novo methods. Such approaches include the design of peptides by iterative growth from an initial position [162] or guiding the growth of the peptide from disemboweled interface residues [163]. Other approaches have been proposed to design peptides based on the linking of fixed elements in the interface with residues [164], property-guided searches [165], ‘docking and linking’ of fragments [166] or optimization [167], including a fast and enhanced search based on ant colony heuristic optimization [168]. A different set of approaches relies on the docking of peptides to a given interface in a dock-and-fold manner [169,170].

## 15. PiPRED: A Knowledge- and Structure-Based Method

### 15.1. The Natural Behavior of Peptide Sequences Can Be Utilized to Design Bioactive Peptides

A different take on modeling and design of peptides based on structure, compared to the de novo approaches described above, is the one utilizing the natural repertoires of peptide conformations, also known as the “knowledge-based” Method. There are advantages of the knowledge-based methods compared to de novo methods: they are less computing intensive, exploiting the natural repertoire of peptide sequences, which has been proven to increase the probability of identifying bioactive peptides [171], as well as not being conformationally biased, and being sequence independent; i.e., there is no need for initial sequence information upon which to model and design peptides.

### 15.2. The Entire Peptide–Protein Interface Is Systematically Explored during Modeling in a Knowledge-Based Approach to Find the Optimal Conformations

PiPreD is an example of a knowledge-based approach to modeling and designing peptides [134]. It relies on a library of structural fragments: iMotifs-DB, extracted from the structure of protein–protein or protein–peptide complexes. Besides exploiting the natural repertoire of structural motifs, PiPreD incorporates native elements of the interface with the targets by using disembodied interface residues, called “anchor residues”. During the modeling stage, the entire interface is systematically and comprehensively explored to find suitable conformations in iMotifs-DB, maximizing the number of anchor residues surrogated by modelled peptides. The modeling stage is totally unbiased, and, therefore, neither the preferred conformation of the peptides nor specific regions or the interface are predetermined. PiPreD has been applied successfully to derive peptides to target RAS-mediated interactions [172] and the Fn14-TWEAK complex [173], and to generate a repertoire of peptide inhibitors of the spike protein, S, of SARS-CoV-2 (*vide infra*). PiPreD can be downloaded from http://www.bioinsilico.org/PIPRED or accessed as a web-server at http://galaxy.interactomix.com/tool_runner?tool_id=interactomix_pipred accessed on 1 December 2021).

The PiPreD approach is likely to be applied successfully to the design of peptide inhibitors of viral fusion proteins.

## 16. PeP*I*-Covid19 Database: A Repertoire of Peptides to Target SARS-CoV-2 Binding to ACE2

### 16.1. Macromolecules such as the Soluble ACE2 Receptor for SARS-CoV-2, Computationally Modified ACE2, Mini-Proteins and Aptamers Can Inhibit Viral Infection

The interaction between S and its cellular receptor, ACE2, is key in the infectious cell entry of SARS-CoV-2 [174,175]. The receptor-binding domain (RBD) of the spike protein recognizes the ACE2 receptor with a high affinity; therefore, blocking this interaction is a therapeutic avenue to fight the disease. Indeed, several recent studies have shown that targeting the RBD with soluble ACE2 [176], including computationally modified ACE2 [177], mini-proteins [178], neutralizing antibodies [179,180,181], nanobodies [182] and aptamers [183,184], can prevent infection. In the same manner, peptides can also contribute to the armory of reagents to fight SARS-CoV-2, as shown by the discovery of potent fusion inhibitors [49], or peptides derived from the N-terminus region of ACE2 [185].

### 16.2. The PePI-Covid19 Database Was Established Based on the PiPreD Method to Design Peptides That Target the Receptor-Binding Region of the S Protein

The PeP*I*-Covid19 database [135] represents yet one more effort in the direction of finding novel peptides to block a key process in the invasion of cells by SARS-CoV-2. Since the start of the COVID-19 pandemic, several structures of protein complexes involving ACE2 and RBD [175,186], as well as monoclonal antibodies [179,180] and nanobodies [182] have become available. We leveraged the structural data to design peptides to target the interaction between the RBD of SARS-CoV-2 and ACE2, and SARS-CoV and ACE2, as contrasting examples, as well as the dimerization surface of ACE2 monomers. The peptides were modelled using the PiPreD method described above, and they recapitulate stretches of residues present in the native interface, plus novel and highly diverse conformations surrogating key interactions at the interface. These peptides are freely available through a dedicated web-based repository, the Pep*I*-Covid19 database (http://bioinsilico.org/pepicovid19, accessed on 1 December 2021), providing convenient access to this wealth of information to the scientific community with the view of maximizing its potential impact in the development of novel therapeutic and diagnostic agents. These peptides are being tested currently in our laboratories.

The PeP*I*-Covid19 database provides a range of peptides that may inhibit SARS-CoV-2 binding to its cellular receptor.

## 17. Concluding Remarks

One of the common mechanisms employed by certain viral fusion proteins, the formation of the six-helix bundle, has been exploited in the development of numerous peptide inhibitors that interfere with the formation of this structure. The localization of the peptide inhibitors in the vicinity of the fusion reaction by the addition of a lipid moiety to the peptide appears to further enhance the inhibitory activity. Lipid conjugation may also facilitate the co-endocytosis of the peptides and viruses whose fusion takes place after endocytosis. Computational approaches are used in the design of peptides to target protein interfaces involved in viral membrane fusion. Peptide design methods are based either on the primary structure of proteins or the three-dimensional structure of the proteins. It will be of great interest to apply these methods to develop inhibitors of the fusion of SARS-CoV-2 with cellular membranes. Indeed, peptides can complement existing strategies in the quest for finding novel therapeutic agents to fight COVID-19, including those exploiting the use of more traditional drug entities (i.e., small chemicals) or macro drugs (e.g., monoclonal antibodies, nanobodies or mini proteins). The peptides may have to be further modified to confer resistance to peptidases and to prolong the availability of these inhibitors in the circulation and the lungs. Future research should also be directed towards the delivery of the therapeutic peptides to the sites of viral infection. Many viral infections are transmitted via the respiratory route, including SARS-CoV-2 and the influenza virus. Thus, the delivery by inhalation of free peptides or peptides anchored in liposomal membranes may be an efficient method to reach sites of initial infection [187].

## Figures and Tables

**Figure 1 pathogens-10-01599-f001:**
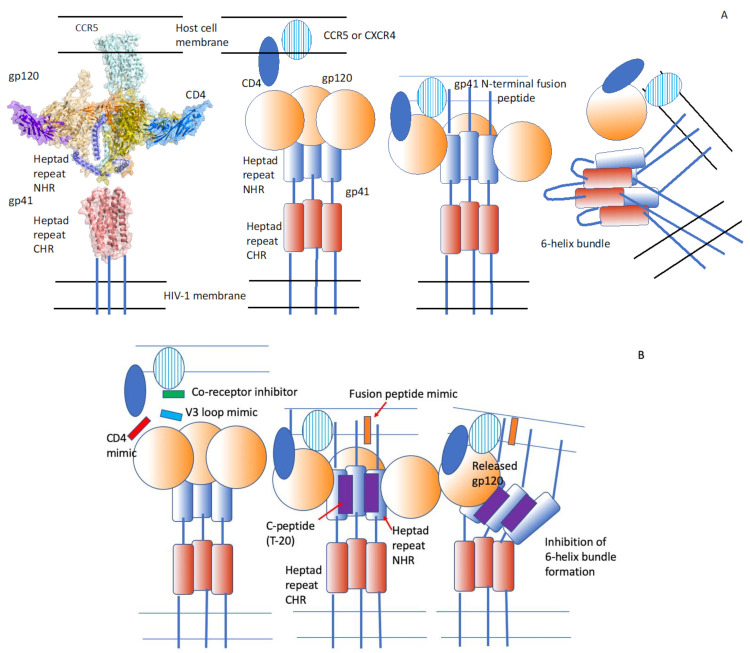
HIV-1 Env-mediated membrane fusion. (**A**) Surface and cartoon representation of the three-dimensional structures of gp120, gp41, CD4 and CCR5. The panel is a composite of the cryo-electron microscopy structure of a full-length gp120 in complex with soluble CD4 and unmodified human CCR5 (PDB code: 6MEO), the structural model of gp120 in complex with CD4 (PDB code: 3J70) and gp41 ectodomain core (PDB code: 1DF5). Binding of gp120 to the CD4 receptor and the co-receptor (a chemokine receptor) and the subsequent conformational changes, leading to the insertion of the gp41 fusion peptide into the host cell membrane, and the formation of the six-helix bundle, comprising the three C-terminal heptad repeat domains and the three N-terminal heptad repeat domains. This conformational change brings the viral and cellular membranes in close proximity to each other. The hydrophobic forces between the membrane-inserted fusion peptides and the membrane anchors of the gp41 are thought to cause membrane fusion. The structural rendering in the left panel was carried out using PyMol (https://pymol.org, accessed on 1 December 2021). (**B**) Inhibition of membrane fusion by various peptides. Red: An inhibitor of gp120-CD4 binding. Green: An inhibitor of gp120 binding to the co-receptor. Blue: An eight-branched peptide mimicking the gp120 V3 loop, thereby inhibiting the interaction of gp120 with the co-receptor. Orange: A peptide similar in composition to the N-terminal fusion peptide of gp41 that competes with the fusion peptide. Purple: A peptide derived from the C-terminal heptad repeat domain of gp41 (CHR) that interacts with the N-terminal heptad repeat (NHR), thereby preventing the normal interaction between the two HR regions. Interaction of gp120 with CD4 and the co-receptor results in the release of the gp120 from the gp41. Modified from Düzgüneş and Konopka [29] (Medical Research Archives 8(9), 1–33; 2020) (https://doi.org/10.18103/mra.v8i9.2244, accessed on 1 December 2021).

**Figure 2 pathogens-10-01599-f002:**
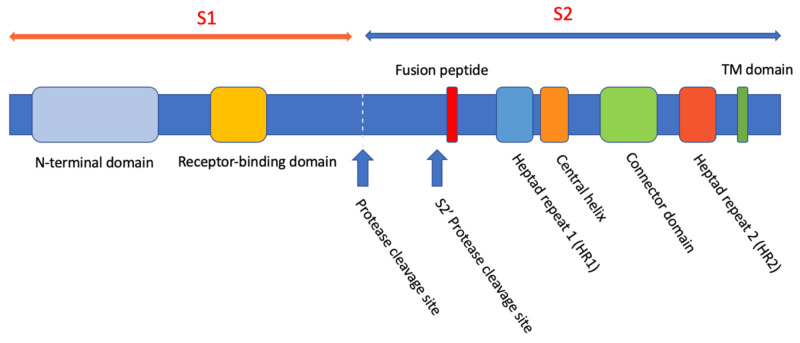
The domains of the SARS-CoV and SARS-CoV-2 spike protein, S. Reproduced from Düzgüneş and Konopka [29] (Medical Research Archives 8(9), 1–33; 2020) (https://doi.org/10.18103/mra.v8i9.2244, accessed on 1 December 2021).

**Figure 3 pathogens-10-01599-f003:**
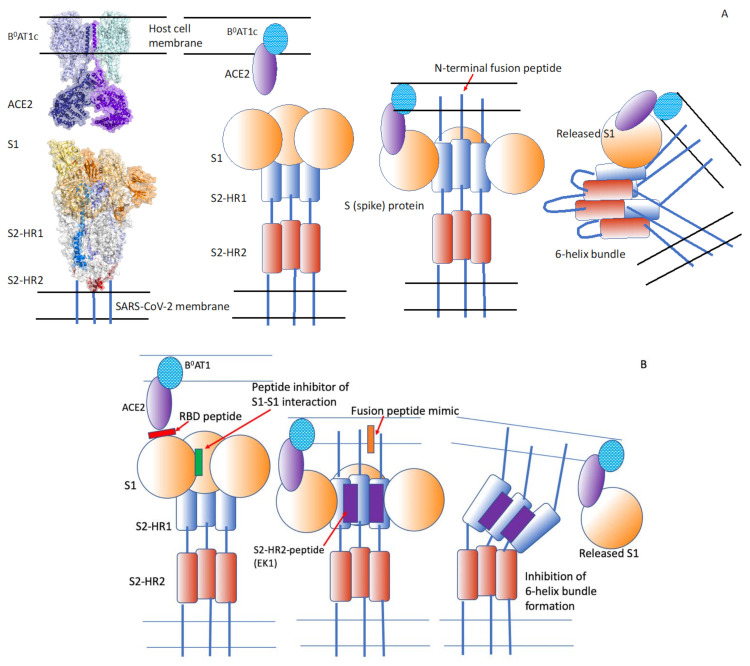
SARS-CoV-2-mediated membrane fusion and its inhibition. (**A**) Surface and cartoon representation of the three-dimensional structures of full-length ACE2, B^o^AT1 and SARS-CoV-2 spike proteins. The left panel is a composite of the cryo-electron microscopy structure of a full-length ACE2 in complex with B^o^AT1 (PDB code: 6m17) and the structure of the trimer SARS-CoV-2 spike protein ectodomain in open form (PDB code: 6VYB). The S1 segment of the spike protein, S, bind to its receptors on host cells. This leads to the insertion of the fusion peptide into the host cell membrane and to the conformational change of the now separate S2 domain of S, resulting in the formation of the six-helix bundle and the close approach of the two membranes. The hydrophobic interactions between the fusion peptide and the transmembrane domains of the S protein leads to membrane destabilization and fusion. The structural rendering in the left panel was carried out using PyMol (https://pymol.org, accessed on 1 December 2021). (**B**) Inhibition of membrane fusion by various peptides. Red: A peptide that binds to the receptor-binding domain of S1. Green: A peptide that inhibits the interaction between S1 subunits. Orange: An S2 fusion peptide mimic that may inhibit the interaction of the fusion peptide with the target membrane. Purple: A peptide derived from the S2-HR2 region that binds with high affinity to the HR1 region and inhibits the interaction between the S2-HR1 and S2-HR2 domains, thus preventing six-helix bundle formation and membrane fusion. Adapted from Düzgüneş and Konopka [29] (Medical Research Archives 8(9), 1–33, 2020 (https://doi.org/10.18103/mra.v8i9.2244, accessed on 1 December 2021).

**Table 1 pathogens-10-01599-t001:** Peptide inhibitors of HIV-1 fusion *.

Peptide	Target	Experimental System	Reference
CBZ-D-Phe-L-Phe-Gly	gp41 fusion peptide	Cell–cell fusion; infection	[18]
gp41 N-terminal hexapeptide	gp41 fusion peptide	gp120/gp41 pseudo-virus fusion	[19]
aa 81–92 of CD4	gp120	Cell–cell fusion; infection	[20]
CD4 CDR-3-like domain	gp120	Cell–cell fusion	[21]
V3 loop eight-branched peptide	gp120	HIV infection	[22]
DP-178	gp41 NHR	Cell–cell fusion	[23,24]
PEG-C34	gp41 NHR	Cell–cell fusion	[25]
Cholesterol-C34	gp41 NHR	Infection	[26]
PIE12-trimer	gp41 NHR hydrophobic pocket	HIV-1 and pseudo-type infection	[27]

* Abbreviation: NHR, N-terminal heptad repeat.

**Table 2 pathogens-10-01599-t002:** Peptide inhibitors of SARS-CoV fusion *.

Peptide	Target	Experimental System	Reference
SARS-CoV HR2 peptide	S-protein NHR (HR1)	Cytopathic effect	[39]
GST-HR2 peptide	S-protein NHR (HR1)	Cytopathic effect	[39]
HR1-1	S-protein CHR (HR2)	Pseudo-typed HIV-luc/SARS virus	[40]
HR2-18	S-protein NHR (HR1)	Pseudo-typed HIV-luc/SARS virus	[40]
CP-1 (from HR2)	S-protein HR1 domain	Infection	[37]
sHR2-8	S-protein HR1 domain	Infection	[41]
P6 (from HR2)	Deep groove of HR1 trimer	S-expressing cell fusion with ACE2-cells	[42]
N46 (from HR1)	S-protein HR2 domain	S-expressing cell fusion with ACE2-cells	[42]
SR9 and SR9EK13	S-protein HR1 domain	Entry at the plasma membrane	[43]

* Abbreviations: NHR (HR1), N-terminal heptad repeat, also termed HR1; GST-HR2, glutathione-S-transferase-HR2 fusion protein; CHR (HR2), C-terminal heptad repeat, also termed HR2.

**Table 3 pathogens-10-01599-t003:** Peptide inhibitors of MERS-CoV fusion *.

Peptide	Target	Experimental System	Reference
HR2P	S-protein NHR (HR1)	Virus replication; cell–cell fusion	[46]
HR2P-M2	S-protein NHR (HR1)	Pseudo-virus infection; cell–cell fusion	[47]
HR2P-M2	S-protein NHR (HR1)	Infection in a murine model	[47]
MERS-5HB	S-protein CHR (HR2)	Pseudo-virus infection; cell–cell fusion	[48]
EK1	S-protein NHR (HR1)	Cell–cell fusion; infection	[49]
EK1C4	S-protein NHR (HR1)	Cell–cell fusion; infection	[49]
SARS-CoV-2 HRC lipopeptide	S-protein NHR (HR1)	Cell–cell fusion; virus infection	[50]

* Abbreviations: NHR (HR1), N-terminal heptad repeat, also termed HR1; CHR (HR2), C-terminal heptad repeat, also termed HR2.

**Table 4 pathogens-10-01599-t004:** Peptide inhibitors of SARS-CoV-2 fusion *.

Peptide	Target	Experimental System	Reference
EK1	S-protein NHR (HR1)	Cell–cell fusion; pseudo-virus infection	[49]
EK1C4	S-protein NHR (HR1)	Cell–cell fusion; pseudo-virus infection	[49]
IBP02	S-protein NHR (HR1)	Cell–cell fusion; pseudo-virus infection	[51]
SARS-CoV-2 HRC lipopeptide	S-protein NHR (HR1)	Cell–cell fusion; virus infection	[50]

* Abbreviations: NHR (HR1), N-terminal heptad repeat, also termed HR1.

**Table 5 pathogens-10-01599-t005:** Peptide inhibitors of influenza virus fusion *.

Peptide	Target	Experimental System	Reference
C18-s2	HA receptor-binding domain	Virus infection	[61]
EB	HA receptor-binding domain	Virus-induced cytopathology	[62]
P155-181-chol	HA internal coiled coil	Virus infection	[64]
43 aa Peptide-CPP	HA2 helix–helix interactions	Virus–liposome fusion; in vivo	[65]
iHA-100	HA stalk domain	Virus infection	[67]
Cyclic P5, P6	HA stem hydrophobic region	Virus neutralization	[68]

* Abbreviations: HA, hemagglutinin; CPP, cell-penetrating peptide.

**Table 6 pathogens-10-01599-t006:** Peptide inhibitors of hepatitis virus binding/fusion *.

Peptide	Target	Experimental System	Reference
Myrcludex B (bulevirtide)	NTCP	Hepatitis B virus infection	[74,76]
E27	E1–E2 protein dimerization	Hepatitis C virus fusion	[78]
CL58	E2	Hepatitis C virus fusion	[79]

* Abbreviations: NTCP, sodium-taurocholate co-transporting polypeptide; E1, E2, Hepatitis C envelope proteins.

**Table 7 pathogens-10-01599-t007:** Peptide inhibitors of flavivirus virus fusion *.

Peptide	Target	Experimental System	Reference
DN57opt	Dengue virus E protein	Virus binding	[112]
1OAN1	Dengue virus E protein	Virus binding	[112]
P5	JEV receptor	Virus infection; tissue pathology	[111]

* Abbreviation: JEV, Japanese encephalitis virus.

**Table 8 pathogens-10-01599-t008:** Peptide inhibitors of herpes virus fusion *.

Peptide	Target	Experimental System	Reference
It1b	HSV-1 membrane	Virus infection	[121]
MelN4	HSV-1 membrane	Virus infection	[121]
gBh1m-Cys-PEG24-Chol	HSV-1 gB protein	Viral cytotoxicity	[115]

* Abbreviation: HSV-1, herpes simplex virus type 1.

**Table 9 pathogens-10-01599-t009:** Peptide inhibitors of filovirus fusion *.

Peptide	Target	Experimental System	Reference
Tat-Ebo	EV GP2 protein	Pseudo-virus and virus infection	[127]
RVFV-6	EV GP2 protein, EV membrane	Virus infection	[128]
Pep-3.2	EV GPcl	EV-GP pseudo-virus	[129]
Pep-3.3	EV GPcl	EV-GP pseudo-virus	[129]

* Abbreviation: EV, Ebola virus type; GPcl, cleaved GP protein of Ebola virus.

## Data Availability

Not applicable.

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
