# Peer review of "Inhibition of Viral Membrane Fusion by Peptides and Approaches to Peptide Design"

_pathogens, 2021, doi:10.3390/pathogens10121599_

Round 1

Reviewer 1 Report

The authors addressed most of my concerns. I agree that detailed structures could obscure overall conformational changes during fusion. However, it is not convincing to me if the schematic diagrams overly simplify the conformations. My recommendation is that improved models can be of interest to the readership. Listing inhibitors in tables under corresponding section can also improve the outline of this review. Overall, this review is useful to some scientists. I am giving advice on how to make is more interesting and accessible to readers.

Author Response

Reviewer 1

The authors addressed most of my concerns. I agree that detailed structures could obscure overall conformational changes during fusion. However, it is not convincing to me if the schematic diagrams overly simplify the conformations. My recommendation is that improved models can be of interest to the readership. Listing inhibitors in tables under corresponding section can also improve the outline of this review. Overall, this review is useful to some scientists. I am giving advice on how to make is more interesting and accessible to readers.

Response

We thank Reviewer 1 for helpful comments.

We have added detailed molecular structures of HIV-1 and SARS-CoV-2 in Figures 1 and 3, respectively. Thus, the simplified segments of the rest of the diagram can be related to the detailed structure.

We think that tables get too cumbersome to follow. The reader can readily go to a section on a particular virus and read about the details of the design of the inhibitory peptides and the results obtained.

We believe the review will be useful to virologists, molecular biologists, pharmaceutical chemists and physicians treating severe viral infections.

Reviewer 2 Report

The review by Duzgunes, covers the many ways in which peptide inhibitors can block viral infection. While parts of it help the reader understand how peptide inhibitors are designed and work, much of the review simply lists every paper that has identified a peptide inhibitor.

Suggestion to improve the readability of the review:

  1. Add topic sentences to the paragraphs. (ie line 73. HIV infection requires the envelope glycoprotein (Env) to both mediate attachment to the host cell and undergo a large conformation change to mediate viral-cellular membrane fusion. And Line 83 – Inhibiting the fusion process is a highly sought after drug target. A number of small peptide inhibitors have been identified that block fusion or entry through various mechanisms. Line 91 – Some peptides can mimic receptor binding and compete for entry.) Without informative topic sentences, it is very difficult to understand the point of each paragraph.
  2. Additionally, the last sentence of each paragraph should summarize the information presented.
  3. The HIV section is written for experts in HIV, not a general virology review. To improve readability I would suggest specifically discussing fewer studies, but give more information on the really important ones that are discussed.
  4. Line 111: What picture may not be straightforward?
  5. As you mention the different types of peptides that inhibit the fusion process, refer to them in the figure.
  6. Summarize the end of each section. (ie HIV – what peptides have been developed that work in cell culture? Which ones are actually in the clinic? Are there problems with them? What is the take home message?)
  7. Does mentioning the specific inhibitory doses of the peptide really matter? What if you focused on how the inhibitory concentration could be improved by membrane targeting? When mentioning the different peptides used for the same virus it would be informative to discuss if they overlap or were different in what specific way.
  8. What about viruses that require endocytosis, how do the peptides end up in the same compartment with the incoming virus at the right concentration? Are there some generalization one can make now that you have gone over all the literature?
  9. Line 382: clinical isolates of measles use both CD150w and nectin 4 which vaccine strains use CD46.
  10. In my opinion, the review should focus on the “methods” and there should be substantial editing on the beginning part. Most of the beginning, virus specific sections, still read as a list of all the peptides that block entry and their concentrations. Filtering this information down to a few take home messages would make a much better review.

Round 2

Reviewer 2 Report

Topic sentences were added, but critical review and important take home messages are still lacking. This is a review for a field expert, not a generalist. 

Author Response

The topic sentences are also the take-home messages. Moreover, we have added summary statements at the end of each main section on each of the virus families.

We did not intend this review only for the general reader. If we had, we would not have generate the 10 tables. A general reader would not necessarily benefit from the detailed list of peptides for each of the virus families, but they might. Moreover, not every specialist, for example in the areas of membrane fusion or virus entry, would have had the time to peruse the literature on 10 different virus groups or families on their own. Thus, our review would also help a specialist gain a broader understanding of the field.

This manuscript is a resubmission of an earlier submission. The following is a list of the peer review reports and author responses from that submission.

Round 1

Reviewer 1 Report

This review on Viral membrane fusion peptide inhibitors covers an interesting topic, yet the manuscript is not ready. The flow of information needs work to enable readers the ability to readily gain information from the review. Below are suggestions to improve the review:

To facilitate reviewers comments please add line numbering.

Bottom of page 1: Suggest breaking up the sentence about the class I fusion proteins because it is confusing adding the localization statement to the first part of the sentence. Maybe add “The tirmers are trafficked to the plasma membrane in infected cells, and are incorporated into the virion membrane following budding” or something like that.

Page 2 top: Typically, 6HB formation is thought to occur at the end of the fusion process, not an intermediate step

Peptide Inhibitors of Viral Membrane Fusion section: While the beginning of the section appears to explore early work looking at how FIP blocks fusion, it then randomly jumps to HR peptide inhibitors. Try to explore the early work and following up with the knowledge about how FIP blocks infection before moving onto HR peptides.

HIV section: This is very choppy, especially the first two paragraphs. There is not clear outline of what the review will cover and simply lists peptide inhibitor studies. Although there is a figure, the figure can be expanded to include the other peptide discussed and more clearly show what effectively inhibits vs what does not. (Can easily decrease size to make it fit better on the page)

Weird font changes in the last HIV paragraph that make it look like it was cut and paste from something else. (This occurs in other places as well)

SARS section: Again very choppy. Reads like an early draft. There are not links between the studies and how they relate to one another.

Figure 3 does not really add anything to figure 1.

In general, the big picture comments above can be used to describe the remaining manuscript. The review primarily reads as a list of the studies that have developed peptide inhibitors for virus fusion, but does not synthesize this work into take home messages.

There are many sentences that are discussing non-fusion related steps in the virus replication cycle that are simply not relevant for this review that can be removed for better flow. Without the line numbers it is too difficult to give specific examples.

Maybe think about reorganizing to discuss the different ways in which peptides can inhibit virus entry and fusion 1) peptides mimicking receptors/receptor binding (competing with receptor or inducing premature triggering). 2) peptides that block fusion by preventing conformational changes (like 6HB). 3) other peptides that block fusion. Rather than listing every study that has developed a HR based peptide, maybe try to go through how new studies built on previous work to make more effective peptides etc.

Reviewer 2 Report

This paper is very well written and very easy to follow. The authors review peptide-derived inhibitors that interfere with forming the common post-fusion conformation - 6-helix bundle of enveloped viruses: HIV-1, SARS-CoV-1, SARS-CoV-2, etc. The reported peptide-derived inhibitors mentioned in this review appear to prevent the virus spike proteins from forming the 6-helix bundle, thus blocking viral membrane fusion.  The fusion models in Fig.1 and Fig.3 over-simplified the structures/conformations of the HIV-1 Env and SARS-CoV-2 S protein. The shapes of spike proteins in these two figures are far from that reported in high-resolution structures. The shedding of the surface subunit from the fusion subunit is very common for Env and S; however, the models in Figs.1 and 3 did not show. I suggest authors consider modifying the fusion models in Figs.1 and 3 to slightly mimic Env and S's reported shapes/structures/conformations. 

The authors also nicely cover different computational methods used for drug design against disease-related functional proteins. I found it very informative. 

Since it is a review, I would suggest authors summarize the pros and cons of each mentioned peptide-design method and provide future directions for increasing ICs and breaths of anti-virus spike peptide-based inhibitors in the concluding remarks.

Some minor issues:
1) There is a Fig.3 duplicate on Page 7.
2) A possible typo in line 2 on Page 14, I assume "ACE2 and RDB should be "ACE2 and RBD".
3) For references regarding conformational changes of virus spike proteins during viral membrane fusion, besides static structures, structural dynamics/conformational dynamics revealed by single-molecule FRET studies should also be cited. The authors should also cite studies of promising anti-HIV-1 Env peptide inhibitors PIE12 and PIE12-cholesterol.

Reviewer 3 Report

The manuscript aims at reviewing peptide inhibitors of viral fusion, with particular emphasis on computational methods for peptide design. The first part, devoted to a number of most important enveloped viruses is a direct recompilation of a recent paper by the same authors published in Medical Research Archives in 2020. This includes also all 3 figures in the manuscript (proper reference is given). Accordingly, this section does not provide any new information beyond what was recently published. The second, much shorter (3 out of 13 pages) part presents methods for the development of protein binding peptides. Information content here is limited: for instance no peptdie-protein docking nor recently popular machine-lerning approaches are mentioned at all. Instead, almost half of this section is devoted to the authors’ own work on (published) PiPred method.  On the editorial side, the text is not clearly written and difficult to follow. Just from the very first pages: if I hadn’t known the differences between class I, II, III fusion proteins or the structure/role of 6-helix bundles, I’d have difficulty to learn about them from the manuscript.  Overall I find it hard to recommend this work.